# THE EXPRESSIVE POWER OF TRANSFORMERS WITH CHAIN OF THOUGHT

**William Merrill**
New York University
`willm@nyu.edu`

**Ashish Sabharwal**
Allen Institute for AI
`ashishs@allenai.org`

## ABSTRACT

Recent theoretical work has identified surprisingly simple reasoning problems, such as checking if two nodes in a graph are connected or simulating finite-state machines, that are provably unsolvable by standard transformers that answer immediately after reading their input. However, in practice, transformers' reasoning can be improved by allowing them to use a "chain of thought" or "scratchpad", i.e., generate and condition on a sequence of intermediate tokens before answering. Motivated by this, we ask: *Does such intermediate generation fundamentally extend the computational power of a decoder-only transformer?* We show that the answer is *yes*, but the amount of increase depends crucially on the amount of intermediate generation. For instance, we find that transformer decoders with a logarithmic number of decoding steps (w.r.t. the input length) push the limits of standard transformers only slightly, while a linear number of decoding steps, assuming projected pre-norm (a slight generalization of standard pre-norm), adds a clear new ability (under standard complexity conjectures): recognizing all regular languages. Our results also imply that linear steps keep transformer decoders within context-sensitive languages, and polynomial steps with generalized pre-norm make them recognize exactly the class of polynomial-time solvable problems—the first exact characterization of a type of transformers in terms of standard complexity classes. Together, this provides a nuanced framework for understanding how the length of a transformer's chain of thought or scratchpad impacts its reasoning power.

## 1 INTRODUCTION

A series of recent theoretical results (Merrill & Sabharwal, 2023b;a; Merrill et al., 2022; Liu et al., 2023; Chiang et al., 2023; Hao et al., 2022) has unveiled surprising limits on realistic formal models of transformers. They have shown that standard transformers, even with ideal parameters, cannot perfectly solve many sequential reasoning problems at scale, such as simulating finite-state machines, deciding whether nodes in a graph are connected, or solving matrix equalities. The intuition here is that the transformer lacks recurrent connections, and recurrence is required to solve these sequential reasoning problems. Empirically, reasoning problems inspired by these results cannot be solved by cutting-edge transformer language models such as ChatGPT and GPT-4 (Zhang et al., 2023), and the reasoning performance of GPT-4 negatively correlates with the depth of the problem's computation graph (Dziri et al., 2023). These results show certain kinds of sequential reasoning pose a challenge for the transformer and motivate extensions to address this issue.

One method that has been empirically successful for improving sequential reasoning with transformers is adding a so-called *chain of thought* (Wei et al., 2022) or *scratchpad* (Nye et al., 2021). These methods allow the transformer to output a sequence of *intermediate tokens* before answering, rather than answering right away after reading the input. Intuitively, such methods could unlock greater expressive power on sequential reasoning problems because the model can use each intermediate token as a kind of recurrent state. Feng et al. (2023) recently showed how chain of thought lets transformers solve a specific modular arithmetic problem that they likely cannot solve without one. Yet there is no general characterization of the class of problems transformers can solve with chain of thought. Thus, the extent to which chain of thought alleviates transformers' weaknesses is unclear, as well as the number of chain of thought steps required to gain reasoning power.

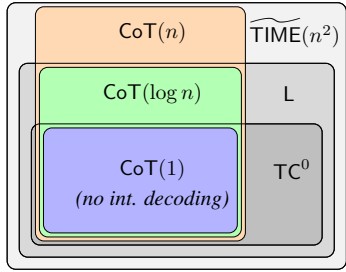 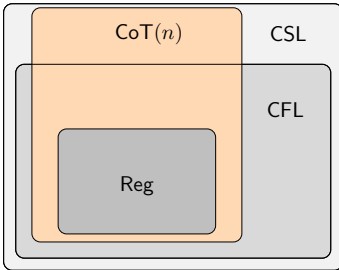

Figure 1: Summary of results: transformers with intermediate generation against various classes of formal languages. A logarithmic number of chain-of-thought steps remains in log-space (L). A linear number of steps adds more power, enabling recognizing all regular languages (Reg), but is contained within context-sensitive languages (CSL). We assume context-free languages (CFL) require $\tilde{\omega}(n^2)$ time to recognize. Some regions with area in the plot are not known to be non-empty.

In this work, we address these open questions by characterizing the reasoning power of transformer decoders that can take *intermediate steps* before generating an answer and comparing them against transformers without intermediate steps. A transformer with a chain of thought constitutes a special case of a transformer decoder with intermediate steps. Our fine-grained results give upper and lower bounds on transformers' power depending on $t(n)$: the number of allowed intermediate steps as a function of the input size $n$. We focus mainly on understanding three regimes: logarithmic steps (when $t(n) = \Theta(\log n)$), linear steps (when $t(n) = \Theta(n)$), and polynomial steps:

1. **Prior Work: No Intermediate Steps.** Recent work has shown transformer decoders without any intermediate steps can only solve problems that lie inside the fairly small circuit complexity class $\mathsf{TC}^0$ (Merrill & Sabharwal, 2023b) and related logical classes (Merrill & Sabharwal, 2023a; Chiang et al., 2023). This implies basic transformers are far from Turing-complete: they cannot even solve problems complete for classes larger than $\mathsf{TC}^0$ such as simulating automata ($\mathsf{NC}^1$-complete), deciding directed graph connectivity (NL-complete), or solving linear equalities (P-complete).[1]

2. **Logarithmic Steps.** With a *logarithmic* number of intermediate steps, we show that the upper bound for transformers expands slightly from $\mathsf{TC}^0$ to L. This means transformers with a logarithmic number of intermediate steps might gain power, but they still cannot solve NL-complete problems like directed graph connectivity or P-complete problems like solving linear equalities.[2]

3. **Linear Steps.** *Linear* intermediate steps allow transformers with projected pre-norm[3] to simulate automata ($\mathsf{NC}^1$-complete), which cannot be done without intermediate steps unless $\mathsf{TC}^0 = \mathsf{NC}^1$. **Polynomial Steps.** With a *polynomial* number of decoding steps, we show that transformers with strict causal attention and projected pre-norm are equivalent to the class P. This, to our best knowledge, is the first equivalence between a class of transformers and a standard complexity class.

Together, our results provide a framework for understanding how the length of a transformer's chain of thought affects its reasoning power. We find a logarithmic chain does not add much, while a linear chain affords more power on inherently sequential reasoning problems.

## 1.1 MAIN RESULTS: POWER OF TRANSFORMERS WITH INTERMEDIATE DECODING

Let $\mathsf{TIME}(t(n))$ be the class of languages $L$ for which there exists a Turing machine that runs in time $O(t(n))$ and accepts $L$.[4] Let $\widetilde{\mathsf{TIME}}(t(n))$ be the class of problems in $\mathsf{TIME}(t(n) \log^k n)$ for some $k$, which is meaningful for $t(n) \geq n$. Let $\mathsf{SPACE}(s(n))$ be the class of languages $L$ for which there exists a Turing machine with tape size bounded by $O(s(n))$ that accepts $L$. We show the following

---

[1] Assuming $\mathsf{NC}^1$, NL, and P do not collapse to $\mathsf{TC}^0$, respectively.

[2] Assuming NL and P do not collapse to L, respectively.

[3] i.e., standard pre-norm but applied only to a *linear projection* of a sublayer's input; cf. Definition 4

[4] As we will define later, this is a non-random-access multitape Turing machine.

relationship between transformers with $t(n)$ steps and standard time/space complexity classes:

$$\mathsf{TIME}(t(n)) \subseteq \mathsf{CoT}(t(n)) \begin{array}{l} \subseteq \mathsf{SPACE}(t(n) + \log n) \\ \subseteq \widetilde{\mathsf{TIME}}(t(n)^2 + n^2) \end{array} . \tag{1}$$

Here $\mathsf{CoT}(t(n))$ denotes the set of languages recognized by some transformer using $t(n)$ decoding steps. Our lower bound (left side of Equation (1)) assumes strict causal saturated attention and projected pre-norm, while upper bounds hold both with and without these architectural assumptions. Both our time lower bound and space upper bound are fairly tight: improving either by a factor larger than $\log t(n)$ would result in a fundamental complexity theory advance (Hopcroft et al., 1977).

**Capabilities of Transformers with CoT.** The left side of Equation (1) implies that transformer decoders with $\Theta(n)$ steps can simulate real-time models of computation like automata or counter machines (Merrill, 2020). Under standard assumptions in complexity theory, transformers with no decoding steps *cannot* simulate all automata (Merrill & Sabharwal, 2023b; Merrill, 2023; Liu et al., 2023). Thus, a linear number of decoding steps makes transformers strictly more powerful. Similarly, the left side of Equation (1) implies transformers with a *quadratic* number of steps can express a linear-time algorithm (for a random access Turing machine) to solve directed graph connectivity (Wigderson, 1992), again a problem known to be beyond the limits of standard transformers. In the same vein, with a *polynomial* number of decoding steps, transformers can solve linear equalities, Horn-clause satisfiability, and universal context-free recognition, all of which are P-complete and thus known to be inexpressible by standard transformers (Merrill & Sabharwal, 2023b).

The left side of Equation (1) is proven by showing transformer decoders can simulate $t$ Turing machine steps with $t$ intermediate steps. Similar prior results have assumed a transformer with external memory (Schuurmans, 2023) or an encoder-decoder model with nonstandard-positional decodings (Pérez et al., 2021). Our construction adapts these ideas to work for a decoder-only model *without* external memory or extra positional encodings, but with strict causal masking and projected pre-norm (cf. Section 2.1).[5] The key idea behind our more general construction is the **layer-norm hash** (Section 3.1): a simple module for effectively storing memory in decoder-only transformers. We believe the layer-norm hash could be broadly useful for building algorithms in transformers. For example, Yao et al. (2021) used a related idea to construct transformers that recognize bounded-depth Dyck languages, although in a more ad hoc way.

**Limitations of Transformers with CoT.** The right side of Equation (1) establishes two upper bounds on transformer decoders with $t(n)$ intermediate steps that depend on both $t(n)$ and $n$. We turn to the implications of this general result in different regimes for $t(n)$:

1. **Log Steps**: Transformer decoders with $O(\log n)$ intermediate steps can only recognize languages in $\mathsf{L} = \mathsf{SPACE}(\log n)$. This implies that transformers with $O(\log n)$ intermediate steps cannot solve NL- or P-complete problems[2] like directed graph connectivity, just like transformers with no intermediate decoding (Merrill & Sabharwal, 2023b).
2. **Linear Steps**: Transformer decoders with $O(n)$ intermediate steps can only recognize languages that are in both $\widetilde{\mathsf{TIME}}(n^2)$ and $\mathsf{SPACE}(n)$. Since $\mathsf{SPACE}(n)$ falls within the context-sensitive languages (Kuroda, 1964), transformers with linear steps can recognize at most context-sensitive languages. Alongside our lower bound, this shows transformer decoders with $\Theta(n)$ steps fall somewhere between regular and context-sensitive languages in the Chomsky hierarchy. Further, transformers with $O(n)$ steps cannot recognize all context-free languages unless context-free languages can be parsed in soft quadratic time.[6]
3. **Polynomial Steps**: If $t(n) = O(n^c)$ for some $c$, we get an upper bound of $\mathsf{P} = \bigcup_{c=1}^{\infty} \mathsf{TIME}(n^c)$. Combined with our lower bound, this shows that transformer decoders with a polynomial number of steps recognize *exactly* the class $\mathsf{P}$. Thus, a polynomial number of steps turns transformers into strong reasoners, though running a polynomial number of forward passes with a large transformer is likely intractable in practice.

Together, these results show that intermediate generation like chain of thought or scratchpad can add reasoning power to transformers and that the number of steps matters as a computational resource

---

[5]Our construction (Theorem 2) can be easily modified to work with an encoder-decoder model as well.

[6]The best known algorithms for context-free recognition run in time $O(n^\omega)$, where $\omega$ is the matrix multiplication constant (Valiant, 1975); the best lower bounds for context-free parsing are sub-quadratic (Lee, 2002).

akin to time or space. Some of the limitations identified in prior work (Merrill & Sabharwal, 2023b; Chiang et al., 2023, etc.) can be overcome with a linear or quadratic number of steps, and a polynomial number of steps covers all problems in P. On the other hand, we have not identified any concrete reasoning problem where a logarithmic number of steps would help. These results provide a unified understanding of the power of transformer decoders across decoding lengths and problems.

## 2 PRELIMINARIES

We study the power of decoder-only transformers that can generate intermediate tokens between reading the input and generating an answer. On input $x \in \Sigma^n$, the transformer consumes tokens $x_1, \ldots, x_n$ for the first $n$ steps, and then, for $t(n)$ *intermediate steps*, consumes the token generated by the previous step. At each step, the transformer can attend over all previous hidden states. This standard method of generating text from a decoder-only model can be described formally as follows. Let $\Sigma$ be a finite alphabet and $f : \Sigma^* \to \Sigma$ be a function mapping a prefix to a next token (parameterized by a transformer). Let $\cdot$ be concatenation. We define the $k$-step extension of $f$ as

$$f^0(x) = x, \qquad f^{k+1}(x) = f^k(x) \cdot f(f^k(x)).$$

We say we have run $f$ on $x$ with $t(n)$ (additional) decoding steps if we compute the function $f^{t(|x|)}(x)$. We consider $f$ with $t(n)$ steps to recognize the language of strings such that $f^{t(|x|)}(x) = 1$, where $1 \in \Sigma$ is a special "accept" symbol. We denote by $\mathsf{CoT}(t(n))$ the set of languages that are recognized by $t(n)$ decoding steps for some transformer $f$.

### 2.1 TRANSFORMERS

A transformer is a neural network parameterizing a function $\Sigma^* \to \Sigma$. Let $\mathbb{D}_p$ be the datatype of $p$-precision floats and define $p$-truncated addition $(+, \sum)$, multiplication $(\cdot)$, and division $(/)$ over $\mathbb{D}_p$ as in Merrill & Sabharwal (2023b). We now define the high-level structure of the transformer in terms of its core components, with the details of those components in Appendix A.

**Definition 1** (Merrill & Sabharwal 2023a). A $p$-precision decoder-only transformer with $h$ heads, $d$ layers, model dimension $m$ (divisible by $h$), and feedforward width $w$ is specified by:

1. An embedding function $e : \Sigma \times \mathbb{N} \to \mathbb{D}_p^m$ whose form is defined in Appendix A.2;
2. For each $1 \le \ell \le d$ and $1 \le k \le h$, a head similarity function $s_k^\ell : \mathbb{D}_p^m \times \mathbb{D}_p^m \to \mathbb{D}_p$ whose form is defined in Appendix A.3 (and includes projected layer-norm);
3. For each $1 \le \ell \le d$ and $1 \le k \le h$, a head value function $v_k^\ell : \mathbb{D}_p^m \to \mathbb{D}_p^{m/h}$ whose form is defined in Appendix A.3 (and includes projected layer-norm);
4. For each $1 \le \ell \le d$, an activation function $f^\ell : (\mathbb{D}_p^{m/h})^h \times \mathbb{D}_p^m \to \mathbb{D}_p^m$ whose form is defined in Appendix A.4 and implicitly uses the feedforward dimension $w$ (and includes projected layer-norm);
5. An output function $\gamma : \mathbb{D}_p^m \to \Sigma$ parameterized as a linear transformation.

**Definition 2.** We define one decoding step $\Sigma^n \to \Sigma$ with a decoder-only transformer as follows:

1. Embeddings: For $1 \le i \le n$, $\mathbf{h}_i^0 = e(x_i, i)$.
2. Multihead Self Attention: For each layer $1 \le \ell \le d$, we compute $h$ attention heads:

$$\mathbf{a}_{i,k}^\ell = \sum_{j=1}^{c(i)} \frac{s_k^\ell(\mathbf{h}_i^{\ell-1}, \mathbf{h}_j^{\ell-1})}{Z_{i,k}^\ell} \cdot v_k^\ell(\mathbf{h}_j^{\ell-1}), \quad \text{where } Z_{i,k}^\ell = \sum_{j=1}^{c(i)} s_k^\ell(\mathbf{h}_i^{\ell-1}, \mathbf{h}_j^{\ell-1})$$

and $c(i)$ is $i$ for standard causal attention and $i - 1$ for strict causal attention.

3. Activation Block: For $1 \le \ell \le d$, activation block $\ell$ maps the head outputs to $\mathbf{h}^\ell$:

$$\mathbf{h}_i^\ell = f^\ell(\mathbf{a}_{i,1}^\ell, \ldots, \mathbf{a}_{i,h}^\ell, \mathbf{h}_i^{\ell-1}).$$

4. Classifier Head: The transformer output is $\gamma(\mathbf{h}_n^d)$.

These definitions use 1-indexing, but when the input contains a beginning-of-sequence token $\$$ (Theorems 1 and 2), we will use 0-indexing starting at $\$$ in the natural way.

**Transformer Precision.** We consider log-precision transformers (Merrill & Sabharwal, 2023b), i.e., we allow the transformer at most $c \log m$ precision for $m$ decoding steps. As a transformer with intermediate generation runs for $n$ input steps and $t(n)$ intermediate decoding steps, this means we have precision at most $c \log(n + t(n))$. Log precision has been analyzed in prior work (Pérez et al., 2021; Merrill & Sabharwal, 2023b;a) because it gives the transformer just enough precision to represent indexes and sums across different positions. This means it naturally formalizes a bounded-precision transformer that is capable of representing position and computing uniform attention, two important capabilities for constructing algorithms with transformers.

Our lower bound constructions (Theorems 1 and 2) assume the following:

1. **Saturated Attention.** A saturated transformer (Merrill et al., 2021) is an idealized transformer with "averaging hard attention" (Strobl et al., 2024): per head, all attention scores are either 0 or $1/v$ for some $v$. This includes uniform attention ($1/n$ over $n$ tokens) or hard attention as special cases. Following common practice (Pérez et al., 2021; Merrill & Sabharwal, 2023b), we use saturated attention for our lower bound constructions.
2. **Strict Causal Masking.** The formulation of attention in Definition 2 makes the slightly nonstandard assumption that causally masked attention at position $i$ can view tokens at all positions up to $i - 1$ but *not* the current token $i$. This is required in Theorem 2.
3. **Projected Pre-Norm.** Our lower bound constructions require $s_\ell$ and $f_\ell$ in Definition 2 to allow a generalization of standard pre-norm. Normally, a layer-norm is applied to the entire input to each sublayer. We generalize this, allowing each sublayer to apply a linear projection before layer-norm. Crucially, in particular, this enables each layer to pick out a subset of the previous hidden state to apply layer-norm to (cf. Definition 4 in Appendix A.1).

For convenience, our proofs with projected pre-norm use an even more general notion of pre-norm, namely **multi-pre-norm**, which allows each sublayer to take $k$ different projections of its input, apply layer-norm to each, and concatenate (cf. Definition 5 in Appendix A.1). Multi-pre-norm can, however, be simulated by multiple layers of projected pre-norm (see Appendix A.1 for a proof):

**Proposition 1** (Chiang, 2024). *Multi-pre-norm with $k$ norms can be simulated by $k + 1$ projected pre-norm layers.*

## 2.2 MODELS OF COMPUTATION

**Automata.** A deterministic finite-state automaton is a tuple $A = \langle \Sigma, Q, q_0, \delta, F \rangle$ where $\Sigma$ is a finite input vocabulary, $Q$ is a finite set of states containing initial state $q_0$, $\delta$ is a transition function $Q \times \Sigma \to Q$, and $F \subseteq Q$ is a set of final states. $A$ processes an input string $\sigma \in \Sigma^n$ as follows. $A$ starts with state $q_0$ and reads $\sigma$ one token at a time, updating $q_i = \delta(q_{i-1}, \sigma_i)$ until $i = n$. $A$ accepts $\sigma$ if $q_n \in F$ and *rejects* it otherwise. The language recognized by $A$ is the set of strings it accepts.

**Turing Machines.** Adapting the notation of Hopcroft et al. (2001), a multitape Turing machine is a tuple $\langle \Sigma, \Gamma, k, b, Q, q_0, \delta, F \rangle$ where:

1. $\Sigma$ is a finite input vocabulary
2. $\Gamma$ is a finite tape vocabulary with $\Sigma \subseteq \Gamma$
3. $k$ is the number of work tapes
4. $b$ is a blank symbol such that $b \in \Gamma$ and $b \notin \Sigma$
5. $Q$ is a finite set of states containing initial state $q_0$
6. $\delta$ is a transition function $(Q \setminus F) \times \Gamma^{k+2} \to Q \times \Gamma^{k+1} \times \{\pm 1\}^{k+2}$
7. $F \subseteq Q$ is a set of halting states

We define Turing machine computation in the standard way (cf. Appendix B).

## 3 LOWER BOUNDS FOR TRANSFORMER DECODERS

Prior work (Merrill & Sabharwal, 2023a) has established strong upper bounds on the reasoning problems transformers can solve. Specifically, under standard conjectures in complexity, transformers without intermediate decoding cannot recognize all regular languages. In this section, we show some of these shortcomings can be overcome with a suitable number of intermediate decoding steps

(and projected pre-norm). Specifically, a linear number of steps enables simulating an automaton. We also show this can be extended to simulate $t(n)$ Turing machine steps with $t(n)$ decoding steps.

## 3.1 Introducing Layer-Norm Hash

We first introduce a useful building block for our results that we call the *layer-norm hash*. The layer-norm hash is a mechanism that enables retrieval across different columns in the transformer based on query-key matching of numerical values. Exact-match retrieval is trivial when the query $q_i$ and keys $k_1, \ldots k_i$ are items in a finite set: just one-hot encode $q_i$ and $k_j$ and the inner product will be maximized when $q_i$ and $k_j$ match. But this does not work when the keys and values are counts produced by uniform attention, which many transformer algorithms use (Weiss et al., 2021), as the key $q_i/i$ and query $k_j/j$ have *different* denominators. The layer-norm hash helps by transforming $q_i/i$ and $k_j/j$ so hard attention retrieves $j$ s.t. $q_i = k_j$. Let $\mathsf{layer\_norm}(\mathbf{x}) = \frac{\mathbf{x}'}{\|\mathbf{x}'\|}$, where $\mathbf{x}' = \mathbf{x} - \bar{x}$.

**Definition 3** (Layer-norm hash). For $x, y \in \mathbb{R}$, let $\phi(x, y) \triangleq \mathsf{layer\_norm}(x, y, -x, -y)$.

$\phi(x, y)$ is a unit vector in $\mathbb{R}^4$. A key feature is scale invariance, and, in particular, that $\phi(x/i, 1/i)$ is invariant w.r.t. $i$ in the sense that it is only a function of $x$, independent of $i$. Let $\phi_x \triangleq \phi(x, 1)$. Then we have the following properties, whose proof may be found in Appendix C.

**Lemma 1** (Scale invariance). *For any $x \in \mathbb{R}$ and $i \in \mathbb{R}_{>0}$, $\phi(x/i, 1/i) = \phi_x$.*

**Lemma 2** (Equality check via layer-norm hash). *For any $q, k \in \mathbb{R}$, $\phi_q \cdot \phi_k = 1$ if and only if $q = k$.*

In other words, the inner product of these representations of two scalars $q$ and $k$, even if computed at different positions $i$ and $j$, respectively, allows us to check for the equality of $q$ and $k$. We can look up key $q_i/i$ in a sequence of keys $k_1/1, \ldots, k_{i-1}/(i-1)$ by attending with query $\phi(q_i/i, 1/i)$ at position $i$ and key $\phi(k_j/j, 1/j)$ at each $j < i$. By Lemmas 1 and 2 this averages the values at all $j$ such that $q_i = k_j$. The layer-norm hash can also be used to directly compare two values $q_i, k_j$ without removing the denominator by computing $\phi(q_i, 1)$ and $\phi(k_j, 1)$.

## 3.2 Simulating Automata

We can use the layer-norm hash to simulate models of computation like automata or Turing machines with intermediate-generation transformers. To warm up, we first show how to use the layer-norm hash to simulate an automaton (i.e., recognize a regular language) and then extend it in Section 3.3 to show how a transformer can simulate a Turing machine for a bounded number of steps.

**Theorem 1** (Regular language recognition). *For any regular language $L$. there is a decoder-only projected pre-norm transformer with strict causal saturated attention (with or without positional encodings) that, on input $\$x$,[7],[8] checks whether $x \in L$ with $|x| + 1$ decoding steps.*

*Proof.* Let $A$ be a finite-state automaton recognizing $L$. We will simulate one step of $A$ with one transformer decoding step (after first reading $n$ input tokens). We refer to tokens with 0 indexing: $\$$ is token 0, $x_1$ is token 1, etc. At step $i, n \leq i \leq 2n$, we will output a token $q_{i-n}$ encoding the next state of $A$. After printing the final state $q_n$, we use one additional step to output 1 iff $q_n \in F$, the set of final states of $A$. At each token $i > 0$, we compute $1/i$ by attending uniformly over the strict left context with value $\mathbb{1}[x_j = \$]$. We show by induction that at step $i \geq n$, we can output $q_{i-n}$.

Base Case: $i = n$. For $i \leq n$, we output $q_0$. Crucially, at $i = n$, this becomes the next input.

Inductive Case: $i > n$. We already have a sequence of intermediate tokens $q_0, \ldots, q_{i-n-1}$. Our goal is to compute $q_{i-n} = \delta(q_{i-n-1}, \sigma_{i-n})$, which first involves retrieving $q_{i-n-1}$ and $\sigma_{i-n}$. $q_{i-n-1}$ is the input to the current column of the transformer. We will use hard attention to retrieve the current input symbol $\sigma_{i-n}$. To do this, we attend uniformly over the prior decoding tokens and $\$$, with a value of 1 at $\$$ and 0 elsewhere. At tokens $i > n$ (i.e., decoding tokens), this yields $\frac{1}{i-n}$. Recall that projected pre-norm can simulate multi-pre-norm (Proposition 1). We now leverage the multi-pre-norm architecture to pass two layer-norms to a feedforward network:

$$\phi_i^{\mathrm{I}} \triangleq \phi(1/i, 1), \qquad \phi_i^{\mathrm{D}} \triangleq \phi(1/(i-n), 1).$$

---

[7]Theorem 1 goes through without strict causal masking (but Theorem 2 will require strict masking).

[8]Theorems 1 and 2 both go through without $\$$ as long as token $j$ can compute value $v_j = \mathbb{1}[j = 0]$.

Let $d_i \triangleq \mathbb{1}[x_i \in Q]$, where $Q$ is the set of states of $A$. Based on $d_i$, we select between $\phi_i^{\mathsf{I}}$ and $\phi_i^{\mathsf{D}}$:

$$\phi_i \triangleq \mathsf{ReLU}(-d_i\vec{1} + \phi_i^{\mathsf{I}}) + \mathsf{ReLU}(d_i\vec{1} - \vec{1} + \phi_i^{\mathsf{D}}).$$

We attend with query layer_norm$(\phi_i) = \phi_i$, key layer_norm$(\phi_j) = \phi_j$ if $d_j = 0$ and $\vec{0}$ otherwise, and value $\sigma_j$ if $d_j = 0$ and $\vec{0}$ otherwise. By Lemma 2, at the current step $i$, the attention score is maximized when $j = i-n$, thus retrieving $\sigma_{i-n}$. We now have the previous state $q_{i-n-1}$ and current token $\sigma_{i-n}$. We conclude by computing $q_{i-n} = \delta(q_{i-n-1}, \sigma_{i-n})$ with a feedforward network. $\square$

Theorem 1 shows that a linear number of decoding steps gives additional reasoning power to log-precision transformers with projected pre-norm (assuming $\mathsf{TC}^0 \neq \mathsf{NC}^1$). This follows because log-precision transformers with no decoding steps are contained in uniform $\mathsf{TC}^0$ (Merrill & Sabharwal, 2023b), which means they cannot recognize all regular languages. In contrast, Theorem 1 says a linear number of steps is sufficient for recognizing all regular languages, establishing a conditional separation. This is an example of simple and familiar additional computational power granted by additional decoding steps. The core challenge in simulating an automaton is recurrence, which cannot be done without decoding steps (Merrill & Sabharwal, 2023b). A linear number of decoding steps allows simulating recurrence, which is where the additional power comes from. However, this added power does not stop with finite-state machines: the layer-norm hash can be used to simulate more complex models of computation such as Turing machines, which we will turn to next.

### 3.3 Simulating Turing Machines

We now show how a transformer decoder can simulate a Turing machine in real time using the layer-norm hash. Our decoder-only construction resembles the encoder-decoder construction of Pérez et al. (2021). However, it avoids simplifying assumptions from Pérez et al. (2021). In addition to assuming non-standard attention and no layer-norm, they required $1/i, 1/i^2$, and $i$ in the positional embeddings, which is problematic because transformers cannot represent unbounded scalars like $i$ due to layer-norm. In contrast, our construction works with or without positional encodings. However, it assumes strict causal masking and projected pre-norm (Section 2.1).

**Theorem 2** (Turing machine simulation). *Let $M$ be a Turing machine that, on input length $1 + n$, runs for at most $t(n)$ steps (at most polynomial). There is a decoder-only projected pre-norm transformer with strict causal saturated attention (with or without positional encodings) that, on input $\$x$,[8] takes $t(n)$ decoding steps and then, with $|M(x)|$ additional steps, outputs $M(x)$.*

*Proof.* We construct a transformer decoder that uses a single decoding step to simulate each Turing machine step. The main difficulty is representing a Turing machine tape in a sequence of transformer state vectors so that the transformer can always correctly reconstruct the value on the tape at the current head position. The key idea will be to store "diffs" to the tape at each step and use the layer-norm hash to dynamically reconstruct the contents at the head position at future steps. Concretely, let $\Delta$ be a finite vocabulary representing elements of $Q \times \Gamma^{k+1} \times \{0, \pm1\}^{k+2}$. The deterministic Turing machine run induces a *diff sequence* $\delta_0, \ldots, \delta_{t(n)} \in \Delta$ capturing the state entered, tokens written, and directions moved after each token. Following the proof of Theorem 1, we use 0-indexing starting at the $\$$ token and compute $1/i$ at each token $i > 0$ as a representation of position. We show by induction that at step $i \geq n$, we can output $\delta_{i-n}$.

Base Case: $i = n$. At every input token (crucially, the last one), we output $\delta_0 = \langle q_0, b^{k+1}, 0^{k+2} \rangle$.

Inductive Case: $i > n$. We first reconstruct $h_i^\tau$, the current position on each tape $\tau$. For each $\tau$, a head attends with query 1, key $\mathbb{1}[x_j \notin \Sigma]$, and value being the move direction of $\tau$ at $j$. Since we assume strict causal attention (for reasons that will become clear later), head $\tau$ thus computes $h_{i-1}^\tau/i$. Since we need $h_i^\tau$, we write both $(h_{i-1}^\tau \pm 1)/i$ to the residual stream. When we need $h_i^\tau/i$ going forward, we use a linear layer to select either $(h_{i-1}^\tau + 1)/i$ or $(h_{i-1}^\tau - 1)/i$ depending on if the current input $\delta_{i-n-1}$ contains a $+1$ move or a $-1$ move for $\tau$, respectively.

We now use two layers to compute the contents at $h_i^0$ on the input tape. Similar to Theorem 1, we use a feedforward network to implement the following piecewise comparison:

$$\phi_i^0 \triangleq \begin{cases} \phi(1, 1/i) = \phi(i, 1) & \text{if } x_i \in \Sigma \\ \phi(h_i^0/i, 1/i) = \phi(h_i^0, 1) & \text{otherwise.} \end{cases}$$

With some abuse of notation, let $\langle \cdot \rangle$ denote vector concatenation. We attend with query $\langle \phi_i^0, -1 \rangle$, key $\langle \phi_j^0, \mathbb{1}[x_j \notin \Sigma] \rangle$, and value $\langle \phi_j^0, \sigma_j \rangle$.[9] Let $\langle \bar{\phi}^0, \bar{\sigma} \rangle$ be the head output. We show in Appendix D that two properties hold. First, by Lemma 3, $\bar{\phi}^0 = \phi_i^0$ iff $1 \le h_i^0 \le n$. Second, by Lemma 4, if $1 \le h_i^0 \le n$, then $\bar{\sigma} = \sigma_{h_i}$. Based on this, we compute the value read from the input tape as $\gamma_i^0 = \bar{\sigma}$ if $\bar{\phi}^0 = \phi_i^0$ and as $\gamma_i^0 = b$ otherwise.

We now use a single attention layer to compute $\gamma_i^\tau$, the contents at $h_i^\tau$ on each non-input tape $\tau$. The layer uses two layer-norm hashes, again taking advantage of the multi-pre-norm architecture that projected pre-norm can simulate (Proposition 1):

$$\phi_i^\tau \triangleq \phi(h_i^\tau / i, 1/i) = \phi(h_i^\tau, 1)$$
$$\psi_i^\tau \triangleq \phi(f(i), 1),$$

where $f(i)$ is defined in Definition 7 in Appendix E. Crucially, $f(i)$ is computable with a single transformer layer and monotonically decreasing with $i$. With strict causal masking, we attend with query $\langle \phi_i^\tau, e_1 \rangle$, key $\langle \phi_j^\tau, -\psi_j^\tau \rangle$, and value $\langle \phi_j^\tau, \delta_{j-n-1} \rangle$. Let $\langle \bar{\phi}^\tau, \bar{\delta} \rangle$ be the head output. We show in Appendix E that two properties hold. First, by Lemma 6, $\bar{\phi}^\tau = \phi_i^\tau$ iff there is some $j < i$ s.t. $h_i^\tau = h_j^\tau$. Second, by Lemma 7, if there is some $j < i$ s.t. $h_i^\tau = h_j^\tau$, then the head retrieves $\langle \phi_j^\tau, \delta_j \rangle$ for the greatest such $j$. Based on this, we compute the last-written value on tape $\tau$ at $h_i^\tau$ as $\gamma_i^\tau = [\bar{\delta}]_{2+\tau}$ if $\bar{\phi}^\tau = \phi_i^\tau$ and $\gamma_i^\tau = b$ otherwise. Having obtained all arguments for the transition function, we now compute $\delta_{i-n} = \delta(q_{i-n-1}, \sigma_{h_i^0}, \gamma_i^1, \ldots \gamma_i^{k+1})$ with a feedforward net.

Finally, we use $|M(x)|$ steps to write the Turing machine output. We detect we are at an output step if either some $\delta_j$ token to the left or the current input encodes a halting state. At each such token $i$, we compute $h_i^{k+1} / i$ as before (recall that tape $k+1$ is the output tape) via attention, except the value now is $d_i^{k+1}$ if $x_i \in \Delta$ and $+1$ otherwise. We attend as before using $h_i^{k+1} / i$ to retrieve (and output) $\gamma_i^{k+1}$. Thus, the $|M(x)|$ tokens generated after a final state are precisely $M(x)$. $\qquad\square$

The critical role projected pre-norm or multi-pre-norm play in Theorems 1 and 2 suggest it could be interesting to investigate incorporating these generalized pre-norms into transformers in practice.

Theorem 2 gives us a general result connecting the power of transformer decoders with $t(n)$ steps to Turing machines with the same number of intermediate steps:

**Corollary 2.1.** $\mathsf{TIME}(t(n)) \subseteq \mathsf{CoT}(t(n))$.

Thus, simulating an automaton (cf. Theorem 1) is not the only new capability unlocked with $\mathrm{O}(n)$ decoding steps: rather, such transformers can solve *any* problem a Turing machine can solve in $\mathrm{O}(n)$ time, such as simulating real-time counter machines (Weiss et al., 2018). With $\mathrm{O}(n^2)$ steps, we can solve directed graph connectivity using standard graph traversal algorithms like depth-first search. Depth-first search runs in $\mathrm{O}(n)$ time on a random access Turing machine (Wigderson, 1992), which can be simulated in $\mathrm{O}(n^2)$ time without random access. Possibly, transformers can solve directed graph connectivity with fewer than $\mathrm{O}(n^2)$ steps, as results from Zhang et al. (2023) hint at.

## 4    UPPER BOUNDS FOR TRANSFORMER DECODERS

Having shown lower bounds on transformers with $t(n)$ steps, we present two different upper bounds: one that relates transformer decoders to time complexity classes, and one that relates them to space complexity classes. The relative strength of the two different bounds will vary depending on $t(n)$. A simple upper bound on transformers with chain of thought can be obtained based on the fact that transformers can be simulated using a quadratic number of arithmetic operations.

**Theorem 3.** $\mathsf{CoT}(t(n)) \subseteq \widetilde{\mathsf{TIME}}(n^2 + t(n)^2)$.

*Proof.* We sketch a multitape Turing machine that will simulate the transformer. Each forward pass $i$ appends key $i$ onto a work tape and value $i$ onto another work tape. To simulate the forward pass at time $i$, it suffices to show how to simulate computing self-attention at time $i$. To compute self

---

[9]As in Theorem 1, a second layer_norm gets applied to $\phi_i^0$ at the start of the layer but has no effect.

attention, the Turing machine first computes the query at time $i$. It then iterates over pairs on the key and value work tapes. For each pair $j$, we compute the attention score between query $i$ and key $j$ and then multiply it by value $j$ using additional work tapes. We then add this value to a running sum tape. We treat the final sum at the output of the attention mechanism.

This runs $n + t(n)$ forward passes, and each forward pass loops over $n + t(n)$ key-value pairs. This means we run at most $\mathrm{O}(n^2 + t(n)^2)$ inner loop calls. It remains to be shown that one inner loop runs in polylogarithmic time. An inner loop just involves adding and multiplying $\mathrm{O}(\log n)$-bit numbers. $p$-bit numbers can be added in time $\mathrm{O}(p) = \mathrm{O}(\log n)$. Similarly, $p$-bit numbers can be multiplied in time $\mathrm{O}(p \log p) \leq \mathrm{O}(p^2)$, which comes out to $\mathrm{O}(\log^2(n + t(n)))$ with log precision. Thus, one inner loop can be run in polylogarithmic time. We conclude that a transformer decoder with $t(n)$ intermediate steps can be simulated by a multitape Turing machine in time $\widetilde{\mathrm{O}}(n^2 + t(n)^2)$. $\square$

Our second upper bound relies on the $\mathsf{TC}^0$ upper bound for transformers without intermediate steps.

**Theorem 4.** $\mathsf{CoT}(t(n)) \subseteq \mathsf{SPACE}(t(n) + \log n)$.

*Proof.* Since log-precision transformers can be simulated in uniform $\mathsf{TC}^0$ (Merrill & Sabharwal, 2023b), they can be simulated in L, i.e., with at most $c \log n$ space overhead on inputs of size $n$. To compute $t(n)$ intermediate decoding steps of a transformer, we store a buffer of at most $t(n)$ generated tokens, which has size $\mathrm{O}(t(n))$. To compute the next token, we call the transformer with an input of size $\mathrm{O}(n + t(n))$ using at most $c \log(n + t(n))$ space overhead. We then clear the memory used and append the finite token generated to the input buffer. It follows from this algorithm that

$$\mathsf{CoT}(t(n)) \subseteq \mathsf{SPACE}(t(n) + c \log(n + t(n)))$$
$$\subseteq \mathsf{SPACE}(t(n) + \log n). \qquad \square$$

With at least $\Omega(\log n)$ steps, this upper bound can be simplified to $\mathsf{SPACE}(t(n))$. The $t(n) = \Theta(n)$ case establishes the context-sensitive languages as an upper bound for transformers with linear steps. Given our $\mathsf{TIME}(t(n))$ lower bound (Theorem 2), the tightest possible space upper bound without making fundamental complexity advances would be $\mathsf{SPACE}(t(n)/\log t(n))$ (Hopcroft et al., 1977). Conversely, our lower bound can only be tightened to $\mathsf{TIME}(t(n) \log t(n))$.

On the other hand, with only $\mathrm{O}(\log n)$ decoding steps, intermediate decoding does not increase expressive power much beyond $\mathsf{TC}^0$, because the upper bound simplifies to $\mathsf{SPACE}(t(n)) = \mathsf{L}$. Thus, under standard assumptions, transformers with a logarithmic number of decoding steps cannot solve directed graph connectivity, Horn formula satisfiability, or other NL- or P-complete problems. Yet, they may be able to solve L-complete problems, unlike transformers without decoding steps.

## 5 CONCLUSION

We have shown that intermediate decoding steps extend the formal power of transformers well beyond previously known upper bounds, such as $\mathsf{TC}^0$ circuits and $\mathsf{FO}(\mathsf{M})$ logic, on transformers without intermediate decoding. Further, the amount of additional power is closely related to the number of decoding steps. In particular, transformers with a linear number of decoding steps have the capacity to recognize regular languages, but cannot recognize languages beyond context-sensitive. With a log number of decoding steps, such transformers can only recognize languages in L, which is a complexity class relatively close to $\mathsf{TC}^0$. Thus, it appears that a linear number of intermediate decoding steps may be required to overcome the limitations of transformers on many sequential reasoning problems of interest. In future work, it may be possible to derive a strict separation between transformers with a log and a linear number of decoding steps and show that certain problems that currently have a quadratic bound can in fact be solved with a roughly linear number of steps.

We have focused on expressive power, rather than analyzing learnability. Whereas our upper bounds directly reveal limitations on what transformers with intermediate generation can learn, one caveat is that our lower bounds *do not* directly imply transformers can learn to use intermediate steps effectively. It would be interesting to formally investigate transformers with CoT from a learning-theoretic lens, possibly along the lines of Malach (2023), and how different kinds of fine-tuning, such as reinforcement learning, might better allow models to use the power of chain of thought.

## ACKNOWLEDGEMENTS

We thank David Chiang for the valuable feedback and for identifying a mismatch between the transformer definition in an earlier version of this paper and standard pre-norm transformers. We also appreciate helpful comments from Gabriel Faria, Ofir Press, Abulhair Saparov, Jason Wei, and Avi Wigderson, as well as researchers in ML2 at NYU and at AI2. WM was supported by NSF Award 1922658, an NSF Graduate Research Fellowship, and AI2.

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

## A  TRANSFORMER COMPONENTS

This section formally defines our generalizations of pre-norm and then recalls the definition from Merrill & Sabharwal (2023a) for the components of the transformer layer.

### A.1  GENERALIZED PRE-NORM

We assume a pre-norm (Xiong et al., 2020) parameterization of the transformer for concreteness, as this is more standard in newer transformers. As stated in Section 2.1, we allow transformer sublayers to apply a linear projection before layer-norm. Concretely, we define $\mathsf{proj\_layer\_norm}(\mathbf{v})$ as follows:

**Definition 4** (Projected pre-norm). Let $\mathbf{M} : \mathbb{R}^m \to \mathbb{R}^m$ be a parameter matrix that projects $m$-dimensional vectors to $m$-dimensional vectors.

$$\mathsf{proj\_layer\_norm}(\mathbf{v}; \mathbf{M}) = \mathsf{layer\_norm}(\mathbf{M}\mathbf{v}).$$

We will omit the parameter $\mathbf{M}$ for convenience, instead writing $\mathsf{proj\_layer\_norm}(\mathbf{v})$.

Our proofs also allow multiple pre-norms of different projections of the hidden state for our lower-bound constructions. Concretely, we define $\mathsf{multi\_layer\_norm}(\mathbf{v})$ as follows.

**Definition 5** (Multi-pre-norm). Let $m$ be divisible by $k$. For $1 \leq i \leq k$, let $\mathbf{M}_i : \mathbb{R}^m \to \mathbb{R}^{m/k}$ be a parameter matrix that projects $m$-dimensional vectors to $m/k$-dimensional vectors. Let $\langle \cdot \rangle_{i=1}^k$ denote iterated concatenation. The *multi-pre-norm* of $\mathbf{v} \in \mathbb{R}^m$ is defined as

$$\mathsf{multi\_layer\_norm}(\mathbf{v}; \mathbf{M}_1, \dots \mathbf{M}_k) = \langle \mathsf{proj\_layer\_norm}(\mathbf{v}; \mathbf{M}_i) \rangle_{i=1}^k .$$

As for projected pre-norm, we will omit the parameters $\mathbf{M}_1, \dots \mathbf{M}_k$ for multi-pre-norm, instead writing $\mathsf{multi\_layer\_norm}(\mathbf{v})$.

As noted earlier, projected pre-norm can simulate multi-pre-norm:

**Proposition 1** (Chiang, 2024). *Multi-pre-norm with $k$ norms can be simulated by $k + 1$ projected pre-norm layers.*

*Proof.* We will simulate a multi-pre-norm layer that takes as input

$$\mathsf{layer\_norm}(\mathbf{M}_1\mathbf{v}), \dots, \mathsf{layer\_norm}(\mathbf{M}_k\mathbf{v})$$

using only projected pre-norm layers. The idea is to use $k$ different projected pre-norm layers (one for each input layer norm). At layer $i$, the layer takes as input $\mathsf{layer\_norm}(\mathbf{M}_i\mathbf{v})$ and writes this to the residual stream. Then, after these $k$ layers, a final layer reads as input

$$\mathsf{layer\_norm}\left( \langle \mathsf{layer\_norm}(\mathbf{M}_i\mathbf{v}) \rangle_{i=1}^k \right) .$$

Since each vector in the concatenation has unit norm, this is equivalent to

$$\frac{1}{\sqrt{k}} \langle \mathsf{layer\_norm}(\mathbf{M}_i\mathbf{v}) \rangle_{i=1}^k .$$

It follows that this layer receives essentially the same input as the original multi-pre-norm layer, up to a constant factor. The original weights for the layer can be multiplied by $\sqrt{k}$ to implement the same computation as the original layer. To make sure that $\sqrt{k}$ is exactly representable, we can pad the number of entries so that $k$ is a perfect square. $\square$

## A.2 Transformer Embeddings

For each position $1 \leq i \leq n$, the transformer embedding function represents token $\sigma_i \in \Sigma$ and its position $i$ with a vector. Let $\mathbf{V}$ be an embedding matrix of size $|\Sigma| \times m$ where each row represents the embedding for some $\sigma$. Let $f : \mathbb{N} \to \mathbb{D}_p^m$ be computable in time $\mathrm{O}(\log n)$. Then,

$$e(\sigma_i, i) = \mathbf{v}_{\sigma_i} + f(i).$$

## A.3 Self Attention

The two components of the self attention block are $s$, the similarity function, and $v$, the value function. Let $\mathbf{h}_i$ be the hidden state at the previous layer and $\bar{\mathbf{h}}_i = \mathsf{multi\_layer\_norm}(\mathbf{h}_i)$. We define similarity of keys and queries as follows:

$$s(\mathbf{h}_i, \mathbf{h}_j) = \exp\left( \frac{\mathbf{q}_i^\top \mathbf{k}_i}{\sqrt{m/h}} \right), \qquad \text{where} \quad \begin{matrix} \mathbf{q}_i = \mathbf{W}_q \bar{\mathbf{h}}_i + \mathbf{b}_q \\ \mathbf{k}_i = \mathbf{W}_k \bar{\mathbf{h}}_i + \mathbf{b}_k \end{matrix} .$$

Then the value function is defined $v(\mathbf{h}_i) = \mathbf{W}_h \bar{\mathbf{h}}_i + \mathbf{b}_h$.

## A.4 Activation Block

The activation function $f$ encapsulates the aggregation of the attention head outputs and the feedforward subnetwork of the transformer. $f$ takes as input attention head outputs $\mathbf{a}_{i,1}, \dots, \mathbf{a}_{i,h} \in \mathbb{D}_p^{m/h}$ and the previous layer value $\mathbf{h}_i$.

The first part of the activation block simulates the pooling part of the self-attention sublayer. The head outputs are first concatenated to form a vector $\mathbf{a}_i$, which is then passed through an affine

transformation $(\mathbf{W}_o, \mathbf{b}_o) : \mathbb{D}_p^m \to \mathbb{D}_p^m$ followed by residual connections to form the sublayer output $\mathbf{o}_i \in \mathbb{D}_p^m$:

$$\mathbf{o}_i = \mathbf{W}_o \mathbf{a}_i + \mathbf{b}_o + \mathbf{h}_i.$$

The second part of the activation block first applies multi-layer-norm and then simulates the feed-forward subnetwork to compute the next layer vector $\mathbf{h}_i'$. Let $\bar{\mathbf{o}}_i = \mathsf{multi\_layer\_norm}(\mathbf{o}_i)$. Let $\sigma$ be a nonlinearity computable in linear time on its input (in the most standard transformer, ReLU). Then, for affine transformations $(\mathbf{W}_1, \mathbf{b}_1) : \mathbb{D}_p^m \to \mathbb{D}_p^w$ and $(\mathbf{W}_2, \mathbf{b}_2) : \mathbb{D}_p^w \to \mathbb{D}_p^m$, the feedforward subnetwork can be defined as:

$$\mathbf{h}_i' = \mathbf{W}_2 \sigma(\mathbf{W}_1 \bar{\mathbf{o}}_i + \mathbf{b}_1) + \mathbf{b}_2 + \mathbf{o}_i.$$

# B   TURING MACHINES

A Turing machine takes as input a string $\sigma \in \Sigma^*$. A *configuration* of a Turing machine is a finite state $q$ along with the contents of an *input* tape $c^0$, $k$ work tapes $c^1, \ldots, c^k$, and an output tape $c^{k+1}$. Finally, for each tape $\tau$, a configuration specifies a head position $h^\tau$. We start with the initial state $q_0$ and the input tape $c_0^0$ containing $\sigma$ starting at position 0 with infinite $b$'s on each side, and $h_0^0 = 0$. All other tapes start containing all $b$'s and with their head at 0. At each time step $i$, if $q_i \notin F$, we recurrently update the configuration by first computing:

$$\langle q_{i+1}, \gamma_i^1, \ldots, \gamma_i^{k+1}, d_i^0, \ldots, d_i^{k+1} \rangle = \delta(q_i, c_i^0[h_i^0], \ldots, c_i^{k+1}[h_i^{k+1}]).$$

We then update tape $\tau$ by setting $c_{i+1}^\tau[h_i^j] = \gamma_i^j$ and keeping all other tape cells the same. We update the head position on tape $\tau$ according to $h_{i+1}^\tau = h_i^\tau + d_i^\tau$. On the other hand, if $q_i \in F$, the Turing machine halts and *outputs* the string of tokens on the output tape from the current head position on the left up to (but not including) the first $b$ on the right. A Turing machine can also be viewed as a language recognizer if we set $\Sigma = \{0, 1\}$ and check if the first output token is 0 or 1.

# C   LAYER-NORM HASH

**Lemma 1** (Scale invariance). *For any $x \in \mathbb{R}$ and $i \in \mathbb{R}_{>0}$, $\phi(x/i, 1/i) = \phi_x$.*

*Proof.* Let $\mathbf{v}_x = \langle x/i, 1/i, -x/i, -1/i \rangle$. $\mathbf{v}_x$ is constructed with mean 0, so layer-norm reduces to RMS-norm (Zhang & Sennrich, 2019). Thus,

$$\phi(x/i, 1/i) = \mathbf{v}_x / \|\mathbf{v}_x\|$$
$$= \mathbf{v}_x \cdot \frac{i}{\sqrt{2x^2 + 2}}$$
$$= \frac{1}{\sqrt{2x^2 + 2}} \langle x, 1, -x, -1 \rangle$$
$$= \phi(x, 1)$$

which, by definition, is $\phi_x$. $\qquad\square$

**Lemma 2** (Equality check via layer-norm hash). *For any $q, k \in \mathbb{R}$, $\phi_q \cdot \phi_k = 1$ if and only if $q = k$.*

*Proof.* By the definition of layer-norm hash, we have

$$\phi(q, 1) \cdot \phi(k, 1) = \frac{1}{\sqrt{2q^2 + 2}} \langle q, 1, -q, -1 \rangle \cdot \frac{1}{\sqrt{2k^2 + 2}} \langle k, 1, -k, -1 \rangle$$
$$= \frac{2qk + 2}{\sqrt{(2q^2 + 2)(2k^2 + 2)}}$$
$$= \frac{qk + 1}{\sqrt{(q^2 + 1)(k^2 + 1)}}.$$

The inner product of unit-norm vectors is maximized at 1. In this case, we show that it achieves 1 only when $q = k$, meaning that is the unique maximum:

$$1 = \frac{qk + 1}{\sqrt{(q^2 + 1)(k^2 + 1)}}$$
$$(qk + 1)^2 = (q^2 + 1)(k^2 + 1)$$
$$(qk)^2 + 2qk + 1 = (qk)^2 + q^2 + k^2 + 1$$
$$2qk = q^2 + k^2$$
$$0 = (q - k)^2.$$

We conclude that $\phi_q \cdot \phi_k$ is maximized (to 1) if and only if $q = k$. $\qquad\square$

As the layer-norm hash is constructed to have mean 0, it does not require a fully general layer-norm implementation and can, in fact, be implemented with simplified RMS norm (Zhang & Sennrich, 2019).

## D    INPUT TAPE RETRIEVAL VIA THE LAYER-NORM HASH

We now describe an attention head that uses the layer-norm hash to read from the input tape in Theorem 2. Define a sequence $h_1, \ldots, h_{i-1}$, which represents Turing machine tape position in Theorem 2.

We define the following layer-norm hash based quantity, which is instantiated in Theorem 2 in a particular way:

$$\phi_i = \begin{cases} \phi(i, 1) & \text{if } 1 \le i \le n \\ \phi(h_i, 1) & \text{otherwise.} \end{cases}$$

The attention head we construct can then be described as follows:

- Query: $\langle \phi_i, -1 \rangle$
- Key: $\langle \phi_j, \mathbb{1}[n < j] \rangle$
- Value: $\mathbf{v}_j \triangleq \langle \phi_j, \sigma_j \rangle$

Let $\bar{\mathbf{v}} \triangleq \langle \bar\phi, \bar\sigma \rangle$ be the head output. This head obeys the following properties:

**Lemma 3.** *Let $i > n$. Then, $1 \le h_i \le n$ if and only if $\bar\phi = \phi_i$.*

*Proof.* We proceed by considering the two directions.

Forward Direction. The query-key inner product has two terms $\kappa^1_{ij} + \kappa^2_{ij}$. By Lemma 2, $\kappa^1_{ij}$ is maximized either when $h^0_i = j$ (and $1 \le j \le n$) or when $h_i = h_j$ (and $n < j$). However, if $n < j$, the second term $\kappa^2_{ij} = 1$. Thus, the attention score is maximized uniquely when $j = h_i$, so the head retrieves $\bar{\mathbf{v}} = \langle \phi(h_i, 1), \sigma_{h_i} \rangle$. Thus, $\bar\phi = \phi(h_1, 1) = \phi_i$.

Backward Direction. We establish bidirectionality by proving the contrapositive. Assume that either $h_i < 1$ or $h_i > n$. The head retrieves $\bar\phi = \frac{1}{|M|} \sum_{j \in M} \phi(j, 1)$ for some $M \subseteq \{1, \ldots, n\}$. It holds that, for all $1 \le j \le n$, $h_i < j$, or the other way around (i.e., for all $1 \le j \le n$, $h_i > j$). Thus, by Lemma 5, $\bar\phi \ne \phi(h_i, 1) = \phi_i$. $\qquad\square$

The following property also emerges from the proof of the forward direction in Lemma 3:

**Lemma 4.** *Let $i > n$. Then, if $1 \le h_i \le n$, $\bar\sigma = \sigma_{h_i}$.*

The backward direction in Lemma 3 relies on the following lemma:

**Lemma 5.** *Let $q \in \mathbb{Z}$ and $k_j \in \mathbb{Z}$ for $1 \le j \le m$. Let $\succ \in \{<, >\}$. If, for all $j$, $q \succ k_j$, then*

$$\phi_q \ne \frac{1}{m} \sum_{j=1}^{m} \phi_{k_j}.$$

*Proof.* Recall that $\phi_x = \phi(x, 1) \in \mathbb{R}^4$ has first element $x/\sqrt{2x^2 + 2}$, which we will denote as $z_x$. Observe that $z_x$ is a monotonically increasing function of $x \in \mathbb{R}$. Thus, $z_q \succ z_{k_j}$ for $1 \le j \le m$, which implies $z_q \succ \frac{1}{m} \sum_{j=1}^{m} z_{k_j}$, from which the lemma conclusion follows. $\qquad\square$

## E   RIGHTMOST RETRIEVAL VIA THE LAYER-NORM HASH

We now describe an attention head that can attend to the rightmost token satisfying a certain property, capturing the construction in Theorem 2 to retrieve the most recent write to a Turing machine work tape. Define a sequence $h_1, \ldots, h_{i-1}$, which represents Turing machine tape position in Theorem 2. As is natural for Turing machine tapes, we assume that if $h \ne h_i$ for all $i$, then it must be that $h \prec h_i$ for all $i$, where $\prec$ is fixed as either $>$ or $<$.

Let $f(i)$ be a tie-breaking term that we will define later in Appendix E.1. We define two layer-norm hash quantities:

$$\phi_i \triangleq \phi(h_1/i, 1/i)$$
$$\psi_i \triangleq \phi(f(i), 1).$$

Recall that $e_1 = \langle 1, 0, 0, 0 \rangle$. Construct an attention head as follows:

- Query: $\langle \phi_i, e_1 \rangle$
- Key: $\langle \phi_j, -\psi_j \rangle$
- Value: $\mathbf{v}_j \triangleq \langle \phi_j, \delta_j \rangle$

Let $\bar{\mathbf{v}} \triangleq \langle \bar{\phi}, \bar{\delta} \rangle$ be the head output. The following properties hold for such a head:

**Lemma 6.** *There exists $j < i$ such that $h_i = h_j$ if and only if $\bar{\phi} = \phi_i$.*

*Proof.* We proceed by considering the two directions.

Forward Direction. The query-key inner product has two terms $\kappa_{ij}^1 + \kappa_{ij}^2$. By Lemma 2, the first term $\kappa_{ij}^1$ is maximized at 1 for each $j$ such that $h_i = h_j$. For $h_i \ne h_j$, $\kappa_{ij}^1 < 1 - 1/(2i^4)$ by Lemma 8. The second component $\kappa_{ij}^2$ monotonically increases with $j$ and satisfies $\kappa_{ij}^2 < f(i) < 1/(2i^4)$ by Lemma 10. Thus, the attention score is maximized for the largest $j < i$ such that $h_i^\tau = h_j^\tau$. Thus, $\bar{\mathbf{v}} = \mathbf{v}_j$ and $\bar{\phi} = \phi_j$ for this $j$, which means $\bar{\phi} = \phi_i$.

Backward Direction. We establish bidirectionality by proving the contrapositive. Assume there is no $j < i$ such that $h_i = h_j$. Then $\bar{\phi} = \frac{1}{|M|} \sum_{j \in M} \phi(h_j, 1)$ for some $M \subseteq \{1, \ldots, n\}$. By assumption (top of Appendix E), we have $h_j \prec h_i$ for all $j < i$. It follows from Lemma 5 that $\bar{\phi} \ne \phi(h_i, 1) = \phi_i$, completing the proof. $\qquad\square$

The following property also emerges from the forward direction of the proof above:

**Lemma 7.** *If there exists $j < i$ such that $h_i = h_j$, then $\bar{\delta} = \delta_j$ for the greatest such $j$.*

### E.1   TIE-BREAKING TERM

The construction above uses a tie-breaker that favors retrieving tokens further to the right. We will justify the construction of such a tie-breaking term here. To begin, we will establish a bound on the layer-norm hash inner product similarity for inexact matches.

**Lemma 8.** *For any $i \ge 2$, $\phi(i, 1) \cdot \phi(i - 1, 1) \le 1 - 1/(2i^4)$.*

*Proof.* Consider the squared dot product:

$$\left(\phi(i,1) \cdot \phi(i-1,1)\right)^2 = \frac{\left(\langle i, 1, -i, -1\rangle \cdot \langle i-1, 1, -(i-1), -1\rangle\right)^2}{(2i^2 + 2)(2(i-1)^2 + 2)}$$

$$= \frac{\left(i(i-1)+1\right)^2}{(i^2+1)((i-1)^2+1)}$$

$$= \frac{i^2(i-1)^2 + 2i(i-1) + 1}{i^2(i-1)^2 + i^2 + (i-1)^2 + 1}$$

$$= \frac{i^2(i-1)^2 + 2i(i-1) + 1}{i^2(i-1)^2 + 2i(i-1) + 2}$$

$$= 1 - \frac{1}{i^2(i-1)^2 + 2i(i-1) + 2}$$

$$= 1 - \frac{1}{i^4 - 2i^3 + 3i^2 - 2i + 2}$$

Since $(1 - y/2)^2 \geq 1 - y$ for any $y$, we have $\sqrt{1-y} \leq 1 - y/2$ for any $y \leq 1$. Applying this to the right hand side of the above equation, we obtain:

$$\phi(i,1) \cdot \phi(i-1,1) = \sqrt{1 - \frac{1}{i^4 - 2i^3 + 3i^2 - 2i + 2}}$$

$$\leq 1 - \frac{1}{2i^4 - 4i^3 + 6i^2 - 4i + 4}$$

$$\leq 1 - \frac{1}{2i^4} \quad \text{for } i \geq 2,$$

which completes the proof. $\qquad\square$

To break ties in attention, we aim to construct a function of $i$ that is computable in the transformer, monotonically decreasing with $i$, and smaller than $1/(2i^4)$. The following definition will accomplish this:

**Definition 6.** We define the following inductively:

$$f(i,0) = 1/i$$
$$f(i,k+1) = f(i-1,k) - f(i,k).$$

By construction, $f(i,k)$ is monotonically increasing and a linear combination of $1/i, 1/(i-1), \ldots, 1/(i-k)$. The latter property means it is computable by a single multihead self-attention layer. To do this, we construct $k$ heads in parallel, where head $h$ attends to all tokens besides the first $h$ and puts value 1 at token $h+1$ and 0 elsewhere.[10] Head $h$ thus computes $1/(i-h)$. We use the linear transformation at the end of the layer to compute $f(i,k)$ via a linear combination of the head outputs.

**Lemma 9.** *For any $k$ and $i > k$, we have*

$$f(i,k) = \frac{k!}{\prod_{j=0}^{k}(i-j)}.$$

*Proof.* By induction over $k$.

Base Case: $i = 0$. We have $f(i,0) = 0!/i$.

---

[10]This head can be implemented by setting a flag in the previous layer at each $i$ for whether $i \leq h$ by hardcoding a comparison between $\phi(1, 1/i)$ and $\phi(h, 1)$.

Inductive Case. We analyze the form of $f(i, k+1)$:

$$f(i, k+1) = f(i-1, k) - f(i, k)$$

$$= \frac{k!}{\prod_{j=0}^{k}(i-1-j)} - \frac{k!}{\prod_{j=0}^{k}(i-j)} \qquad \text{(Inductive assumption)}$$

$$= k! \cdot \frac{\prod_{j=0}^{k}(i-j) - \prod_{j=0}^{k}(i-1-j)}{i \prod_{j=1}^{k}(i-j)^2(i-k-1)} \qquad \text{(Form common denominator)}$$

$$= k! \cdot \frac{i \prod_{j=1}^{k}(i-j) - (i-k-1) \prod_{j=1}^{k}(i-j)}{i \prod_{j=1}^{k}(i-j)^2(i-k-1)} \qquad \text{(Pull out factors)}$$

$$= k! \cdot \frac{(k+1) \prod_{j=1}^{k}(i-j)}{i \prod_{j=1}^{k}(i-j)^2(i-k-1)} \qquad \text{(Distributive property)}$$

$$= \frac{(k+1)!}{i \prod_{j=1}^{k}(i-j)(i-k-1)} \qquad \text{(Simplify)}$$

$$= \frac{(k+1)!}{\prod_{j=0}^{k+1}(i-j)}. \qquad \square$$

It remains to be shown that $f(i, k)$ can be made smaller than $1/(2i^4)$. To handle edge cases around small values of $i$, we define:

**Definition 7.** Let $\epsilon = 10^{-10}$. For $i \geq 1$, let

$$f(i) = \begin{cases} 1/1000 - \epsilon i & \text{if } i \leq 4 \\ f(i, 3)/100 & \text{if } i \geq 5 \end{cases}$$

**Lemma 10.** *For $i \geq 1$, $f(i) < 1/(2i^4)$.*

*Proof.* When $i \leq 4$, we have $f(i) < 1/1000 < 1/(2i^4)$.

When $i \geq 5$, by Lemma 9, we have:

$$\frac{f(i, 3)}{100} = \frac{3!}{100i(i-1)(i-2)(i-3)}$$

It can be verified that the values of $i$ for which this expression equals $1/(2i^4)$ are all in the interval $[0, 5)$, and that for $i \geq 5$, $(100/6)i(i-1)(i-2)(i-3) > 2i^4$. $\qquad \square$

