# OpenReview forum: "The Expressive Power of Transformers with Chain of Thought"
_ICLR.cc/2024/Conference — ICLR 2024 poster_

### Official Review · Reviewer_7d6w · 2023-10-31

**Soundness:** 3 good
**Presentation:** 3 good
**Contribution:** 2 fair
**Rating:** 6
**Confidence:** 4

**Summary:**

This submission studies theoretical expressive power of transformer models equipped with chain of thought reasoning. The authors provide theoretical lower and upper bounds for different decoding step sizez, log (L), linear (Time n^2; Space n), and polynomial (P). Their proof for the lower bounds relies on a construction that enables retrieval across different columns in the transformer (“Layer-norm hash”), which is used to simulate automata and Turing machines in transformer forward passes. The upper bounds are proven by straightforwardly simulating attention via a TM.

**Strengths:**

- Interesting theoretical work on the expressive power of transformers that sheds light on the limitations (and possible capabilities) of transformers for researchers and practicioners alike
- The theory is well developed and well presented

**Weaknesses:**

- The paper is not very well self-contained and relies on the reader being familiar with two previous papers
- The last paragraph in the conclusion should be clearly marked as limitation of this theoretical bounds (the lower bounds and upper bounds might not have any relecance in practice).

**Questions:**

- For the upper bounds: Doesn’t the same reasoning as for the lower bounds apply? For fixed data distributions, complexity classes are irrelevant. How does this mismatch fit into the theory work?

---

> ### Author Response · Authors · 2023-11-15
>
> Thanks for your review!
>
> 1. **Self-Containedness:** We will add more specific details of the papers we build upon where appropriate. We will also provide a reference to a new comprehensive survey paper on formal language results about transformers (https://arxiv.org/abs/2311.00208).
> 2. **Conclusion:** Regarding the conclusion, we will rephrase the part about lower bounds to clarify that it is a limitation. Namely, that just because there exists a transformer that can implement some behavior, it does not mean that that transformer can be learned by gradient descent.
>
> > For the upper bounds: Doesn’t the same reasoning as for the lower bounds apply?
>
> The reasoning for lower bounds actually does not apply for the upper bounds. The reasoning for the lower bounds is: just because a transformer can express a solution to a problem, it does not mean that gradient descent will find that solution. In the other direction, if we know that a transformer cannot express a solution to a problem, we know that gradient descent will not find a solution to the problem, because no solution exists. We will clarify this.

---

> > ### Comment · Reviewer_7d6w · 2023-11-22
> > **Thanks**
> >
> > I thank the authors for their submission and answer! I would like to keep my score as is. I would appreciate if the authors provide a clear limitation section instead of a sentence in the conclusion. Especially for a more practically oriented audience, the difference between theoretical expressive power and learning in practice might not be clear and need to be made explicit.

---

> > > ### Author Response · Authors · 2023-11-22
> > >
> > > We appreciate your suggestion and plan to add a limitations section if accepted. The most high-priority item to discuss here will be the current caveat paragraph about lower bounds mentioned in the conclusion.

---

### Official Review · Reviewer_a7kJ · 2023-10-31

**Soundness:** 3 good
**Presentation:** 3 good
**Contribution:** 3 good
**Rating:** 8
**Confidence:** 3

**Summary:**

This paper studies the reasoning ability of transformer models, which underlie some state-of-the-art deep neural networks such as ChatGPT or GPT-4, in solving sequential reasoning problems such as simulating finite-state machines, deciding connectivity of two nodes in a graph, and solving matrix equalities, which are formalized as classical language classes (context free grammars or regular languages in the Chomsky hierarchy) or traditional complexity classes (bounded-depth threshold circuits $TC^0$, logarithmic space $L$, polynomial time $P$).

The paper shows that the number of intermediate steps (length of Chain-of-Thought, or amount of scratchpad space) is closely related to the sequential reasoning ability of a decoder-only architecture transformer. In particular, Equation 1 shows that the class of problems solvable with $t(n)$ intermediate steps are those solvable by multitape Turing machines using similar time or space.

**Strengths:**

The connection to traditional complexity classes (Equation 1) effectively refactors out all recurrence/use of scalable memory into the intermediate steps in chain of thought/scrathpad using transformers. This is as expected, and such intuition shows up also in studying traditional complexity classes such as $L$ and $P$ in a natural way, contrasting the need of memory to sequential reasoning.

The proof of this connection (Equation 1) is also natural. In particular, the simulation of Turing machines by decoder-only transformers using an encoding based on layer-norm hash is also natural.

**Weaknesses:**

The study of log-precision transformer models with $h$ heads, $d$ layers, dimension $m$, and feedforward width $w$, while capturing some of the trends in Machine Learning in recent years, may not apply when the trend changes to other non-transformer architectures in a decade or two.

The first equivalence between a transformer-defined classes (the chain of thought class) with a standard complexity class ($P$) is not too unexpected, given that both sides allow for a more lenient notion of equivalence/reductions, namely closed under polynomial time reductions instead of more strict notion of reduction such as logarithmic space reductions. Equating under polynomial time reductions are easier targets to hit.

**Questions:**

While some future questions are mentioned (for example to study how different kinds of fine-tuning such as reinforcement learning or improving a model's ability to use chain of thought), how could such learnability problems be approached at a high level? The current paper studies expressive power by simulation results (between chain-of-thought transformers and Turing machines in both directions), which are somewhat standard in the computer science literature and appears unable to answer learnability questions by far.

---

> ### Author Response · Authors · 2023-11-15
>
> Thank you for your review!
>
> > The study of log-precision transformer models with h heads, l  layers, dimension d, and feedforward width w, while capturing some of the trends in Machine Learning in recent years, may not apply when the trend changes to other non-transformer architectures in a decade or two.
>
> We agree that this is perhaps an inherent limitation of theory done about transformers. However, we do believe that, even in the world where transformers become obsolete, there could some insight to be gained by analyzing them that might inform the design of future models and the techniques develops might aid in analyzing future models.
>
> > The first equivalence between a transformer-defined classes (the chain of thought class) with a standard complexity class (P) is not too unexpected, given that both sides allow for a more lenient notion of equivalence/reductions, namely closed under polynomial time reductions instead of more strict notion of reduction such as logarithmic space reductions.
>
> Could you clarify what you mean by stricter reductions in this context? The conversion from multitape TMs to transformers is tight (TIME(t) in CoT(t)), and the conversion from transformers to multitape TMs has a quadratic factor (CoT(t) in soft TIME(t^2)).
>
> > While some future questions are mentioned (for example to study how different kinds of fine-tuning such as reinforcement learning or improving a model's ability to use chain of thought), how could such learnability problems be approached at a high level?
>
> This is a good question. At a very high level, such work would have to define “learnable” by leveraging some condition that makes gradient descent likely to find a solution or some property of transformers that is likely to emerge from gradient descent.
>
> We believe that recent work characterizing the [simplicity biases of transformers](https://arxiv.org/abs/2211.12316) might be relevant here. Other work identifies [restricted classes of transformers that are learned in practice](https://arxiv.org/abs/2010.09697), and perhaps analyzing the expressive power of such a simplified model of transformers could be a first step towards getting at what is learnable vs. what is expressible. More ambitiously, it might be possible to add constraints to the hypothesis class defined in terms of the geometry of the loss landscape (e.g., what problems can be solved such that the loss function is locally very flat). Some other recent work obtains learnability results for neural networks in the [statistical query learning](https://arxiv.org/abs/2207.08799) and next-token predictors in the [PAC learning frameworks](https://arxiv.org/abs/2309.06979), and perhaps that would be another avenue to pursue with transformer language models.

---

> > ### Comment · Reviewer_a7kJ · 2023-11-20
> >
> > > Could you clarify what you mean by stricter reductions in this context?
> >
> > Between these two reductions:
> > 1. logarithmic space reduction
> > 2. polynomial time reduction
> > (1) is a stricter reduction than (2), that is, every logarithmic space reduction is a polynomial time reduction (but not vice versa or else  $L = P$).
> >
> > Note that polynomial time reductions are used in both:
> > a. the definition of $P$, and
> > b. the quadratic factor in Equation 1.
> >
> > So "equating $P$ (which is closed under polynomial time reductions) with some other classes (CoT classes) under polynomial time reductions" is easier than, say, "equating $L$ (which is closed under stricter reductions) with some other CoT classes under logarithmic reductions".

---

### Official Review · Reviewer_XLkW · 2023-11-01

**Soundness:** 3 good
**Presentation:** 3 good
**Contribution:** 3 good
**Rating:** 8
**Confidence:** 3

**Summary:**

A series of recent theoretical work has analysed the limitations of transformers when it comes to solving certain simple sequential problems. In this work, the authors analyse whether these negative results remain when transformers are enriched with a chain-of-thoght. The main results show that these extensions extend the power of transformers to beyond TC^0 and firs-order logic.

**Strengths:**

The theoretical results are interesting. In particular, the authors have proved that transformers with a linear number of decoding steps have the capacity to regocnize regular languages. Nevertheless they cannot recognised all context sensitive languages.

**Weaknesses:**

The paper focus on the analysis of the expressive power of transformers enriched with chain of thought. For the audience of ICLR, it would be interesting to have some results analysing aspects related to learnability of such models.

**Questions:**

No questions.

---

> ### Author Response · Authors · 2023-11-15
>
> Thanks for your review! We appreciate that you found our theoretical results interesting.
>
> We agree that questions around learnability are interesting but they lie outside the scope of this work. For future work, we are thinking about whether it may be possible to say something about learnability in the PAC learning framework by extending ideas from [this recent paper](https://arxiv.org/pdf/2309.06979.pdf).

---

### Official Review · Reviewer_DD5G · 2023-11-07

**Soundness:** 3 good
**Presentation:** 3 good
**Contribution:** 2 fair
**Rating:** 8
**Confidence:** 3

**Summary:**

This paper investigates how the formal reasoning power of decoder-only transformers change if they are allowed to generate intermediate outputs that can be read in subsequent steps, akin to chain-of-thought or scratchpad reasoning.  The authors show that for an input of size n, adding log n intermediate steps doesn't add much to the power of transformers without intermediate steps.  However, adding a linear number of intermediate steps results in a significant increase in reasoning power -- allowing transformers to recognize languages somewhere between regular and contex-sensitive languages in the Chomsky hierarchy.  The authors also show that using t(n) intermediate steps allows a transformer to mimic t(n) steps of a Turing machine.  Finally, they also show that allowing a transformer poly(n) intermediate steps precisely captures the class of problems solvable in deterministic polynomial time.  The paper presents complexity-theoretic lower and upper bounds of the computational power of transformers with the ability to use t(n) intermediate steps, for various functions t(n).

**Strengths:**

The paper establishes both lower and upper bounds on the computational power of transformers with intermediate steps (akin to chain-of-thought or scratchpad reasoning).  The lower and upper bounds are almost tight.  Although I'm not an expert in this area, I find that this work significantly extends state-of-the-art in relating the computational power of transformers to classical complexity classes.  The notion of norm-hash is likely useful in other contexts as well.

**Weaknesses:**

The paper is technically intricate, and may not be accessible to the general ICLR audience.  The core idea of using norm-hash is not illustrated well.  Using examples to explain the intuition will help in improving the readability of the paper.  The paper assumes a lot in terms of prior knowledge of the reader in the formal language theoretic modeling of transformers.  The notation is at times unnecessarily complicated -- I understand this was done for the sake of rigour, but this does hurt readability of the paper.

**Questions:**

1. Please explain the intuition behind norm-hash using examples.  Otherwise, its introduction appears too abrupt.  This is an important concept for the proofs, and hence it should have been explained more clearly with examples.
2. Why is it necessary to consider a multi-tape Turing machine when you want to show that a move of a Turing machine can be simulated by one move of a transformer with intermediate steps?  Wouldn't showing this for a single tape Turing machine suffice?
3. The result concerning characterization of P is not adequately explained in the paper.  For example, there is no subsection devoted to it. It is only mentioned as a bullet on page 3.  Please elaborate on this a bit more, like the other results.
4.  The paper presumes a lot about prior knowledge of the reader in the formal language theoretic modeling of transformers.  A more gentle introduction would help the readability of the paper.

**Details Of Ethics Concerns:**

No ethics concerns.

---

> ### Author Response · Authors · 2023-11-15
>
> Thanks for your review!
>
> # 1) Layer-Norm Hash
>
> We will add an example problem like the following the better illustrate the layer norm hash. Potentially, if space allows, we would like to have a figure illustrating it visually.
>
> **Problem:** Imagine we describe the state of a 1D grid world with x’s and o’s and then specify a sequence of +/- moves in that world: `xoooxo+-+-++-`. We then want a transformer to add up the +/- moves and return whether the grid cell is `x` or `o`.
>
> The natural algorithm is to compute the current position with c = #(`+`) - #(`-`) and then attend to that position and see whether it’s x or o. A transformer with causal masking can compute c/n, but it’s messy to use this to retrieve the grid cell at position c because that cell is represented as 1/c. The layer-norm hash lets implement this hard attention cleanly by attending with query phi(c/n, 1/n) = phi_c and key phi(1/c, 1) = phi_c!
>
> We hope this helps make the idea more concrete, and it captures the way we later use the layer-norm hash for automata and Turing machines in a simpler setting.
>
> # 2) Turing Machine Simulation
>
> Regarding the Turing machine simulation, indeed, you’re correct that it would suffice to simulate a single tape to prove Turing completeness.We consider the multitape model because this is the standard model used to define finegrained complexity classes within P (e.g., TIME(n)). The reason for this is only having just a single tape can add polynomial overhead compared to multitape TM. Thus, if we want to conclude TIME(t) \subseteq CoT(t) as in Corollary 2.1, we need to use the multitape model.
>
> # 3) Clarifying P Result
>
> > The result concerning characterization of P is not adequately explained in the paper. For example, there is no subsection devoted to it.
>
> Thanks for pointing this out. We will add text explaining that this follows from Corollary 2.1 (TIME(t) in CoT(t)) and Theorem 3 (CoT(t) in softTIME(t^2)). Together, these results show poly(n) CoT can express any problem in P and vice versa.
>
> # 4) Reference for Transformers and Formal Languages
>
> Finally, we would like to address your suggestion to give readers more context for formal language results about transformers. Given the space limitations, we think the best way to provide a gentler introduction is to add a reference to a [new survey paper](https://arxiv.org/abs/2311.00208) on transformers and formal language theory.

---

> > ### Comment · Reviewer_DD5G · 2023-11-22
> > **Thanks for your responses**
> >
> > Thanks to the authors for responding to my questions.  My concern about accessibility of the material in this paper to the general ICLR audience remains.  I wish the authors also provided some pointers to practical implications of their results.

---

### Comment · Area_Chair_z8rJ · 2023-11-19
**I would not be interested in this paper.**

While the reviewers liked this paper, I find this area of research irrelevant to ICLR.  There is no reason to believe that this paper has any relevance to practice. Given the reviews I will feel obligated to accept.  But the authors (and other reviewers) should be aware that had I been a reviewer I would have rejected it on the grounds of being off-topic --- the topic being understanding actual transformers.  Sub-communities can reinforce each other get papers accepted even if the sub-community is not doing anything interesting.  So one can always say "but papers of this type have been accepted in the past".  Eventually the area gets more and more parochial and the mainstream community just ignores it.

---

> ### Public Comment · ~Sabri_Eyuboglu1 · 2024-05-11
>
> For what it is worth, I discovered this work at ICLR. I am primarily an empirical researcher and don't consider myself to be part of this theoretical sub-community. Nonetheless, I found this paper very interesting and expect these results could influence my future work and AI practice. I'm glad the paper was accepted :)

---

### Meta-Review · Area_Chair_z8rJ · 2023-12-14

**Metareview:**

This paper gives formal expressivity results for chain of thought transformers.  Deep network expressivity results typically involve the depth of the network.  The classes given here involve the length of the chain of thought rather than the transformer depth.  One result is that with polynomial length thought chains transformers can recognize any polynomial time decidable language.  This result seems natural and exactly what one would expect if we assume a transformer can express a state-transition table.

The main strength is that these results are novel (to my knowledge) and natural.

There are two weaknesses.  First, the results are perhaps too natural --- they are just what one would expect.  Second, a result about expressivity and relating expressivity to complexity classes would seem to be of limited interest to the mainstream of ICLR community.
 The relevance to advancing AI practice is very questionable.

**Justification For Why Not Higher Score:**

I believe these results will be of limited interest to the mainstream ICLR community.

**Justification For Why Not Lower Score:**

There is a (small) community interested in these questions and papers contributes some (unsurprising) results.

---

### Decision · Program_Chairs · 2024-01-16

Accept (poster)